# Metabolic and Transcriptomic Analyses Reveal the Effects of Ethephon on *Taraxacum kok-saghyz* Rodin

**DOI:** 10.3390/molecules27113548

**Published:** 2022-05-31

**Authors:** Zhanjiang Zhang, Guang Shen, Yihua Yang, Cui Li, Xiaoying Chen, Xiaonan Yang, Xiaoyun Guo, Jianhua Miao, Li Li, Ming Lei

**Affiliations:** 1Guangxi Key Laboratory for High-Quality Formation and Utilization of Dao-di Herbs, Guangxi Botanical Garden of Medicinal Plants, Nanning 530023, China; zzj1811@163.com; 2Institute of Natural Resources and Ecology, Heilongjiang Academy of Sciences, Harbin 150040, China; shen19772@163.com (G.S.); yyh_1975@163.com (Y.Y.); 3Guangxi Key Laboratory of Medicinal Resources Protection and Genetic Improvement, Guangxi Botanical Garden of Medicinal Plants, Nanning 530023, China; licuicui941@aliyun.com (C.L.); chxying2018@126.com (X.C.); nannangood33@163.com (X.Y.); gxyun2008@sina.com (X.G.); 4Guangxi Engineering Research Center of TCM Resource Intelligent Creation, Guangxi Botanical Garden of Medicinal Plants, Nanning 530023, China; 5School of Modern Industry for Selenium Science and Engineering, National R&D Center for Se-Rich Agricultural Products Processing Technology, Wuhan Polytechnic University, Wuhan 430023, China

**Keywords:** *Taraxacum kok-saghyz* Rodin, transcriptomics, metabolomics

## Abstract

The roots of *Taraxacum kok-saghyz* Rodin (TKS) are well-known and valued for their rubber-producing ability. Therefore, research on the analysis and detection of metabolites from the roots of TKS have been reported in previous studies. However, all of these studies have the shortcoming of focusing on only the rubber of TKS, without profiling the other metabolites in a systematic and comprehensive way. Here, the primary and secondary metabolites from the leaves of TKS were investigated using UPLC–ESI–MS/MS, and a total of 229 metabolites were characterized. Carboxylic acid derivatives, fatty acyls, phenols, and organooxygen compounds were found to be the major metabolites of TKS. The transcriptome data indicated that ribosomal, glycolysis/gluconeogenesis, phenylpropanoid biosynthesis, and linoleic acid metabolism genes were significantly differentially expressed. This study is the first to report the differences in the metabolic and transcriptome profiles of TKS leaves under exogenous ethephon spray, which improves our understanding of the main metabolites and their molecular mechanisms in TKS leaves.

## 1. Introduction

Among certain rubber-producing species, *Taraxacum kok-saghyz* Rodin (TKS) has gained special attention in recent years as a potentially commercially viable rubber plant because its rubber is very similar in quality to *Hevea brasiliensis* rubber [1]. TKS belongs to the composite family; it is native to the Tianshan Mountain regions and a traditional medicinal plant in China [2,3]. TKS, as an annual crop, can be broadly planted in temperate regions, and its roots can produce a large amount of natural rubber (NR) [2]. However, no attention has been paid to the available parts of TKS, besides the roots. In fact, the leaves of TKS may also contain many nutrients that can be developed into dietary supplements and feed ingredients in the future. Therefore, it is of great practical significance to study the metabolites of TKS, in order to promote the development of this industry.

Hormones play important roles in the regulation of seed germination, root elongation, shoot dormancy, and secondary metabolism [4,5]. Ethephon (2-chloroethyl phosphonic acid), an important exogenous hormone, can promote seed germination and rooting [5]. In addition, ethephon is commonly used in the production of tomatoes [6], apples [7], bananas [8,9], grapes [10], citrus [11,12], watermelon [13], and other crops to accelerate ripening. However, the effect of ethephon on rubber grass has not been studied. Here, we used exogenous ethephon to stimulate the primary and secondary metabolisms of TKS at different stages. Then, the transcriptome and metabolome were sequenced in an untreated group (control check group, CK), as well as 2 (Eth2), 4 (Eth4), 8 (Eth8), and 24 h (Eth24) of treatment. All datasets provided important information on the mechanisms of nutrient biosynthesis in TKS regulated by ethephon. We aimed to uncover the differences in the metabolite profile and transcriptome of rubber grass leaves at different stages after exogenous ethephon spraying. This is the first study to evaluate the nutritional value of rubber grass, and our work will guide future research on breeding and resource utilization.

## 2. Results

### 2.1. Overview of the Metabolites and Quality Control Analysis of the TKS Samples

To better understand the effect of exogenous ethephon treatment on the metabolites of TKS at four different time points, the primary and secondary metabolites in the samples were identified by UPLC–MS. A total of 229 metabolites were divided into amino acids, lipids, carbohydrates, nucleotides, cofactors, vitamins, xenobiotics, energy compounds, and other components (Figure 1B). Among them, amino acids had the greatest contribution (28%), followed by lipids (24.02%) and carbohydrates (21.4%), which were the main components in TKS in this study.

Hierarchal clustering of the metabolic profile of TKS was performed during five different periods (0, 2, 4, 8, and 24 h) (Figure 1A), and the results are shown in the heatmap (Figure 1C). There were two main clusters in the pattern of metabolite accumulation, and most of the metabolites showed significant accumulation at 2 h. However, the remaining metabolites significantly decreased during the maturation process. These results indicate that the abundance of some metabolites gradually decreases with increasing exposure time to ethephon. Accordingly, the details of these differentially expressed metabolic pathways were analyzed in this study. Analysis of the differential metabolite pathways in the five periods revealed that four pathways were significantly affected: taurine and hypotaurine metabolism, glyoxylate and dicarboxylate metabolism, galactose metabolism, and phenylalanine metabolism (Figure 1D).

### 2.2. Analysis of the Differential Metabolites

To identify the key factors affecting metabolite composition, we used principal component analysis (PCA) to analyze all of the metabolites (Figure 2A). The PCA components displayed clear separations between the four different periods after exposure to exogenous ethephon, compared with CK, indicating that ethephon has a substantial effect on metabolite composition. Additionally, the six biological replicates of each group were clustered together, indicating good homogeneity between the replicates and high reliability of the data. The metabolite compositions at 8 and 24 h were more similar to each other than those observed in the other groups. The closer relationship of the replicates in the 8 and 24 h groups might be caused by termination of the metabolite changes being reached at 8 h; therefore, these plants have similar metabolism. Additionally, we compared the metabolome data of samples from five periods and found that the concentrations of six compounds ((S)-2,3,4,5-tetrahydropyridine-2-carboxylate, 3-Indoleacetonitrile, betaine aldehyde, anabasine, betaine aldehyde, and pipecolic acid) decreased with time (Appendix A), and the concentrations of four compounds (3-(2-Hydroxyphenyl)propanoic acid, bovinic acid, D-Galacturonate, and gingerol) increased with time (Appendix A).

Compared with CK, 41 metabolites were upregulated and 16 metabolites were downregulated in the Eth2 group (Figure 2B). Among the upregulated metabolites were one pyridine derivative (pyridoxine), one pteridine phenol derivative (riboflavin), four phenols (hydroquinone, norepinephrine, capsaicin, and hydroquinone), one azacyclic compound (D-1-piperideine-2-carboxylic acid), two organonitrogen compounds (betaine aldehyde and choline), one organic phosphoric acid derivative (acetylphosphate), one ajmaline-sarpagine alkaloid (raucaffricine), one hydroxy acid derivative (L-malic acid), three organooxygen compounds (ribitol, sucrose, and l-gulose), three hydroxy acid derivatives (L-malic acid, 4-hydroxycinnamic acid, and m-coumaric acid), three benzene-substituted derivatives (4-hydroxyphenylacetaldehyde, phenylpyruvic acid, and 3-hydroxybenzoic acid), seven carboxylic acid derivatives (2-aminoacrylic acid, pipecolic acid, L-isoleucine, N-formyl-L-methionine, saccharopine, ureidopropionic acid, and (Z)-but-1-ene-1,2,4-tricarboxylate), and five fatty acyls (13S-hydroxyoctadecadienoic acid, succinic acid semialdehyde, 9-(S)-HPODE, 2-isopropylmalic acid, alpha-linolenic acid, and melibiitol). Among the downregulated metabolites were one organic oxide (lactose 6-phosphate), five organooxygen compounds (L-ribulose, mannitol, glyceric acid, threonic acid, and tartaric acid), two hydroxy acids and derivatives (caffeat and trans-ferulic acid), two Benzene-substituted derivatives (4-(beta-D-Glucosyloxy) benzoate and protocatechuic acid), two carboxylic acid derivatives (gamma-aminobutyric acid and isocitric acid), and two fatty acyls (3-methylthiopropionic acid and bovinic acid) were identified.

Fifty-two metabolites were upregulated and 16 metabolites were downregulated in the Eth4 group (Figure 2C). Among the upregulated metabolites were 2 pyrimidine nucleosides (deoxycytidine and deoxyuridine), 2 pyridine derivatives (pyridoxine and nicotinic acid), 1 coumarin derivative (umbelliferone), 3 benzene-substituted derivatives (4-hydroxyphenylacetaldehyde, 4-(beta-D-glucosyloxy)benzoate, and phenylpyruvic acid), 1 indole derivative (3-methylindole), 1 imidazopyrimidine (Adenine), 13 carboxylic acid derivatives (4-methylaminobutyrate, L-malic acid, 2-hydroxy-2-ethylsuccinic acid, 2,3-dinor-8-iso prostaglandin F1 alpha, pipecolic acid, L-glutamic acid, L-isoleucine, L-aspartic acid, cis-4-hydroxy-D-proline, 5-aminopentanoic acid, (Z)-but-1-ene-1,2,4-tricarboxylate, ureidopropionic acid, and 2-O-(alpha-D-mannosyl)-D-glycerate), 7 fatty acyls (oleic acid, prostaglandin F2a, prostaglandin G2, alpha-linolenic acid, 2-isopropylmalic acid, melibiitol, and L-2-hydroxyglutaric acid), 1 phenol ether (estragole), 1 keto acid derivative (D-4-hydroxy-2-oxoglutarate), 2 organonitrogen compounds (sphinganine and glucosamine), 1 phenols (norepinephrine), 2 organooxygen compounds (ribitol and sucrose), 1 endogenous metabolite (D-cathine), 2 cinnamic acid derivatives (m-coumaric acid and 4-hydroxycinnamic acid), 1 ajmaline-sarpagine alkaloid (raucaffricine), and 1 flavonoid (luteolin 7-O-beta-D-glucoside). Among the downregulated metabolites were two benzene-substituted derivatives (chlorogenic acid and protocatechuic acid), one imidazopyrimidine (hypoxanthine), two carboxylic acid derivatives (gamma-aminobutyric acid, isocitric acid, and N-formyl-L-methionine), one fatty acyl (hexadecanedioate), piperidine, one prenol lipid ((S)-2,3-epoxysqualene), six organooxygen compounds (mannitol, tartaric acid, galactaric acid, lactose 6-phosphate, glyceric acid, and threonic acid), and one endogenous metabolite (D-galactose).

Forty metabolites were upregulated and fifty-four metabolites were downregulated in the Eth8 vs. CK comparison (Figure 2D). Among the upregulated metabolites were 6 fatty acyls (5,6-DHET, oleic acid, 15-deoxy-d-12,14-PGJ2, 2-isopropylmalic acid, and melibiitol), 17 carboxylic acid derivatives (L-valine, gamma-glutamyl-beta-aminopropiononitrile, 2,3-dinor-8-iso prostaglandin F1 alpha, 5-aminopentanoic acid, 9,10,13-TriHOME, 2-hydroxy-2-ethylsuccinic acid, ketoleucine, L-glutamic acid, L-aspartic acid, L-malic acid, L-isoleucine, citric acid, ureidopropionic acid, D-altronate, 4-methylaminobutyrate, L-arogenate, and succinic acid), 2 organooxygen compounds (threonic acid and 3D-3,5/4-trihydroxycyclohexane-1,2-dione), 1 carboximidic acid derivative (N-carbamoylputrescine), 1 pyrimidine nucleoside (deoxycytidine), 1 phenol ether (estragole), 1 diazine (thymine), 2 coumarin derivatives (umbelliferone, coumarin), 1 keto acid derivative (D-4-hydroxy-2-oxoglutarate), 3 benzene-substituted derivative (4-hydroxyphenylacetaldehyde, 4-(beta-D-glucosyloxy)benzoate, phenylethylamine), 1 cinnamic acid derivative (4-hydroxycinnamic acid), 2 organonitrogen compounds (choline and glucosamine), 1 indole derivative (5-hydroxyindoleacetic acid), 1 benzene-substituted derivatives (phenylpyruvic acid), 1 ajmaline-sarpagine alkaloid (raucaffricine), and 1 keto acid derivative (oxoadipic acid). Among the downregulated metabolites were five fatty acyls (erucic acid, 13S-hydroxyoctadecadienoic acid, palmitic acid, 9(S)-HPODE, 3-methylthiopropionic acid, and stearic acid), eight carboxylic acid derivatives (4,5-dihydroorotic acid, creatinine, (Z)-but-1-ene-1,2,4-tricarboxylate, 2-aminoacrylic acid, cis-4-hydroxy-D-proline, gamma-aminobutyric acid, isocitric acid, and 2-O-(alpha-D-mannosyl)-D-glycerate), two pyridine derivatives (nicotinic acid and pyridoxine), one pyrrole (pyrrole-2-carboxylic acid), four organooxygen compounds (D-mannose, tartaric acid, lactose 6-phosphate, and glyceric acid), two phenols compounds (hydroquinone and capsaicin), one pyrroline (1-pyrroline-4-hydroxy-2-carboxylate), one pteridine derivative (riboflavin), one pyrimidine nucleoside (uridine), one diazine (Dihydrouracil), one azacyclic compound (4-hydroxyproline), one organic sulfonic acid derivative (taurine), three cinnamic acid derivatives compounds (caffeate, m-coumaric acid, and trans-ferulic acid), one imidazopyrimidine (adenine), one purine nucleoside (inosine), and one endogenous metabolite (D-galactose).

Forty-seven metabolites were upregulated and thirty-nine metabolites were downregulated in the Eth24 vs. CK comparison (Figure 2E). Among the upregulated metabolites were 5 fatty acyls (5,6-DHET, 15-deoxy-d-12,14-PGJ2, bovinic acid, 2-isopropylmalic acid, and melibiitol), 14 carboxylic acid derivatives (L-valine, 4,5-dihydroorotic acid, 2-aminoacrylic acid, 2,3-dinor-8-iso prostaglandin F1 alpha, 5-aminopentanoic acid, 2-hydroxy-2-ethylsuccinic acid, ketoleucine, L-glutamic acid, L-aspartic acid, L-malic acid, L-isoleucine, guanidoacetic acid, L-arogenate, and D-galacturonate), 1 pyridine derivative (pyridoxine), 3 organooxygen compounds (3D-3,5/4-trihydroxycyclohexane-1,2-dione, sucrose, and ribitol), 1 pyrroline (1-pyrroline-4-hydroxy-2-carboxylate), 1 carboxyimide derivative (L-proline), 2 pyrimidine nucleosides (deoxycytidine and uridine), 1 diazine (thymine), 2 diazines (umbelliferone and coumarin), 1 coumarin derivative (D-4-hydroxy-2-oxoglutarate), 1 cinnamic acid derivative (4-hydroxycinnamic acid), 1 organic sulfonic acid derivative (taurine), 1 indoles derivative (5-hydroxyindoleacetic acid), 1 ajmaline-sarpagine (raucaffricine), 1 phenylpropionic acid (3-(2-hydroxyphenyl)), 1 propanoic acid (phenylpyruvic acid), 1 benzene-substituted derivative (phenylpyruvic acid), and 1 phenol (gingerol). Among the downregulated metabolites were five fatty acyls (erucic acid, 9(S)-HPODE, 13S-hydroxyoctadecadienoic acid, linoleic acid, prostaglandin F2a, and 3-methylthiopropionic acid), one pyridine derivative (nicotinic acid), seven organooxygen compounds (D-mannose, L-ribulose, D-glucose-1-phosphate, glyceric acid, mannitol, and threonic acid), one pteridine derivative (riboflavin), indole, one indole derivative (3-methylindole), one diazine (dihydrouracil), one nitrogen heterocyclic compound (4-hydroxyproline), one cinnamic acid derivative (caffeate), one isoprene lipid ((S)-2,3-epoxysqualene), one keto acid derivative (pyruvic acid), one flavonoid (luteolin 7-O-beta-D-glucoside), one purine nucleoside (inosine), and one imidazopyrimidine (adenine).

### 2.3. TKS Transcriptomic Analysis Overview

To research the mechanism of ethephon treatment of TKS, transcriptome analysis was performed on TKS leaves treated with exogenous ethephon. The Eth2, Eth4, Eth8, and Eth24 samples were collected and sequenced. After removing the low-quality reads, a total of 621,257,990 clean reads were obtained. The Q30 scores were >93%, indicating that the quality of the transcriptome sequencing data was high. These transcripts and unigenes were annotated in the gene ontology (GO), KEGG orthology (KO), eukaryotic ortholog group (KOG), non-redundant protein sequence database (NR/NT), and PFAM database. After hierarchical clustering, the longest cluster sequence was obtained as a unigene for subsequent analysis.

A total of 46,734 genes were functionally annotated in the databases. Moreover, 9433 (3841 up- and 5592 downregulated), 7621 (3400 upregulated and 4221 downregulated), 7373 (3705 upregulated and 3668 downregulated), and 8244 (3439 upregulated and 4805 downregulated) differentially expressed genes (DEGs) were identified from the pairwise comparison of CK vs. Eth2, CK vs. Eth4, CK vs. Eth8, and CK vs. Eth24, respectively. These results indicated that ethephon could induce transcript changes in TKS leaves.

To better understand the effect of exogenous ethephon treatment on the transcriptome of TKS, hierarchal clustering of the transcriptome profile at the five different time periods (0, 2, 4, 8, and 24 h) was performed, and the results are shown in the heatmap (Figure 3A). There were two main clusters in the transcriptome accumulation pattern, and most of the transcriptome had significant accumulation at 2 h. We next performed PCA, and in line with the hierarchical clustering results, the samples were divided into five groups. Samples at the same stage had similar transcriptome patterns (Figure 3B). However, the CK and 24 h treatment groups were clustered together in the PCA score plot, indicating that these two transcriptome profiles were similar.

To compare the transcriptome at the five time points, we generated an advanced Venn diagram to show the intersection of the transcriptome at the different times (Figure 3C). In line with the hierarchical clustering results, Eth2 exhibited the greatest transcriptome abundance. In addition, Eth2 and the other samples shared a large number of transcriptome factors. This means that Eth2 has many factors that overlap with CK and the samples at 4, 8, and 24 h. These results showed that Eth2 had the greatest transcriptome abundance, followed by 8 and 24 h, with 4 h having a lower transcriptome abundance.

### 2.4. TKS Differentially Expressed Genes Analysis

KEGG enrichment analysis was performed on all DEGs at the different stages. Four pairwise comparisons of DEG enrichment between each comparison (CK vs. Eth2, CK vs. Eth4, CK vs. Eth8, and CK vs. Eth24) were established, and we found that exogenous ethephon had the greatest impact on 22 to 26 metabolic pathways (Figure 4). Then, all of the DEGs were mapped to the GO database and further categorized into three classifications, including cellular component, molecular function, and biological process.

Volcano plots were constructed to determine the number of transcripts that were significantly changed after inflorescence removal. The significant DEGs met the criteria | Log2 (fold change) | ≥ 1 and FDR ≤ 0.05. A total of 9433 DEGs (3841 upregulated and 5592 downregulated) were identified for CK vs. Eth2 (Figure 5A), and 7343 DEGs (3705 upregulated and 3668 downregulated) were identified for CK vs. Eth4 (Figure 5B). A total of 8 244 DEGs (3439 upregulated and 4805 downregulated) were found for CK vs. Eth8 (Figure 5C). Additionally, 7621 DEGs (3400 upregulated and 4221 downregulated) were found for CK vs. Eth24 (Figure 5D).

To identify the major enriched functional terms that correspond with the DEGs, GO enrichment analysis was carried out separately for the four comparisons (CK vs. Eth2, CK vs. Eth4, CK vs. Eth8, and CK vs. Eth24) (Figure 6). For GO enrichment, the DEGs of CK vs. Eth2, CK vs. Eth4, CK vs. Eth8, and CK vs. Eth24 were significantly enriched in 4049, 3637, 3778, and 3591 GO terms, respectively, which belonged to three major functional categories: cell component (CC), molecular function (MF), and biological process (BP) (Figure 6). For the four pairwise comparisons, the top terms in the CC category were: response to ribosome (GO: 0005840), intrinsic component of plasma membrane (GO: 0031226), and intrinsic component of membrane (GO: 0031224). The top terms in the MF category were: response to structural constituent of ribosome (GO: 0003735), structural constituent of ribosome (GO: 0003735), oxidoreductase activity (GO: 0016491), and catalytic activity (GO: 0003824). The top terms in the BP category were response to carbohydrate metabolic process (GO: 0005975), ribosome biogenesis (GO: 0042254), oxidation−reduction process (GO: 0055114), and carbohydrate metabolic process (GO: 0005975).

## 3. Discussion

Ethylene is an important plant hormone that is thought to be involved in the inductive and signal transduction pathways in plants. Previous studies have shown that ethylene can activate genes, cause the accumulation of various metabolites, and promote the production of phenolic acids in strawberries [14] and carrots [15]. In addition, ethylene can promote an improvement in the medicinal components of medicinal plants, such as increasing the content of salvianolic acid B in *S. miltiorrhiza* hairy roots [16]. Ethephon, one of the most widely used growth regulators, can be converted very quickly to ethylene when sprayed on plants. In this study, the effects of exogenous ethephon spraying at four different time points within 24 h were different. The changes induced by exogenous ethephon after 2 h were the most dramatic, and the results of the metabolomic and transcriptomic analyses showed that the effects of ethephon spray at this time point were significantly different from those at the other four stages.

Here, we found that ethephon can significantly influence the primary metabolites in TKS, such as glucose (1.8 times higher), threonic acid (2.8 times lower), sucrose (2 times higher), and malic acid (3.75 times higher), all of which were affected by exogenous ethephon spraying within 2 h. Interestingly, luteolin 7-O-beta-d-glucoside, which can reduce oxidative stress and inflammatory mechanisms, was detected in the Eth4 group (2 times higher). In the Eth8 and Eth24 groups, carboxylic acid derivatives were the most affected compounds, with 17 and 13 impacted, respectively. In contrast to our results, Jiaqi Kong reported the active ingredients in Taraxacum kok-saghyz Rodin from different places [17]. In the previous study, TKS was found to be rich in pentacyclic triterpenes and sterols. However, we did not detect any pentacyclic triterpenes or sterols in TKS. This may be related to the differences in the detection methods and varieties. Additionally, we found that the content of lysine increased; however, with the extension of time, we did not find that the content of lysine increased at 4, 8, and 24 h. On the contrary, under the action of ethephon, the content of phenylalanine decreased within 2 h, but increased after 8 h. Besides, we also found that six and four compounds were downregulated and upregulated with the induction of ethephon, respectively. This means that the significant increased and decreased compounds, which were time-dependent, may be considered a manifestation of the stress induced by the ethephon. Furthermore, we found that treatment with ethephon resulted in an increase in ribosome, glycolysis/gluconeogenesis, phenylpropanoid biosynthesis, linoleic acid metabolism, starch and sucrose metabolism, and lysine biosynthesis. Additionally, we found the metabolomics and transcripts of TKS were greatly affected after 2 h of ethephon stimulation, and the effect of ethephon at 4, 8, and 24 h was significantly different from 2 h (Figure 1C). Therefore, we have reason to believe that ethephon can have a significant effect on the primary metabolites of TKS in a short period of time.

A previous study collected a time series of the metabolomic and proteomic data from latex-exuding and non-exuding TKS roots [18]. They examined the effects of selectively degrading all laticifer RNA on the chemistry and proteome of a laticifer-containing plant [18]. Here, we present a comprehensive analysis of the metabolome and transcriptome of TKS at five different stages. The results are shown for the detection of 229 metabolites and 46,734 expressed genes. Additionally, we revealed that the main components of TKS leaves were amino acids, lipids, carbohydrates, nucleotides, cofactors, vitamins, xenobiotics, and energy compounds, which differs from previous studies on rubber. We also found that many metabolites, such as carboxylic acid derivatives, fatty acyls, phenols, and organooxygen compounds, were enriched after exogenous ethephon spraying for 24 h. Limitations regarding the types of bioactive substances detected in previous studies have restricted our understanding of the chemical composition of TKS leaves; thus, our results have significantly elucidated the chemical composition of TKS leaves. Moreover, this study provides chemical and RNA-seq information regarding exogenous ethephon spraying on TKS, which can provide helpful guidance for the development of dietary supplements and feed ingredients in the future.

## 4. Materials and Methods

### 4.1. Plant Materials and Elicitor Treatments

All TKS seeds were provided and identified by Konkova Nina, associate researcher, N. I. Vavilov All-Russian Institute of Plant Genetic Resources; the accession number for the germplasm bank in the Russia organization is K445. Plants were cultured in Nanning, Guangxi Province, China, in the middle of July 2020. The TKS seeds were soaked in 0.7% potassium permanganate solution for 2 h. Then, we washed the seeds (2 or 3 seeds in each pot) with purified water and planted them in blow-molded black pots (60 mm diameter · 55 mm height) filled with peat and garden soil (1:1), and the potted plants were placed in a 23 °C incubator. Ethephon was purchased from Shanghai Yuanye Bio-Technology Co., Ltd. (Shanghai, China). The 100 μmol/L of ethephon solution was prepared in distilled water. Then, 100 mL of ethephon (100 μmol/L) was sprayed onto the leaf surface of TKS at the four-leaf stage. Finally, the leaves of untreated and treated TKS were collected at 0 (untreated group, CK), 2, 4, 8, and 24 h (Figure 1A). These samples were flash-frozen in liquid nitrogen containers and stored at −80 °C for further analysis.

### 4.2. Metabolite Extraction and UPLC–MS Sample Preparation

The leaves of TKS were freeze-dried, homogenized with glass beads in a grinder at 1500 rpm for 1 min, and ultrasonicated at room temperature for 15 min. After that, the powdered samples (200 mg) of each sample were weighed into a 2 mL EP tube and extracted with 600 μL of methanol (−20 °C). The whole target isotope internal standard (Succinic acid-2,2,3,3,-d4; Cholic acid-2,2,3,4,4,-d5; DL-Tryptophan-2,3,3-d3; DL-Methionine-3,3,4,4-d4; L-Phenylalanine(Ring-D5); Choline chloride (trimethyl-d9)) was added, followed by vortexing for 30 s. Then, each mixture was centrifuged at 16,260× *g* for 10 min, 300 μL of the supernatant was filtered through a 0.22 μm membrane, and the filtrate was added to chromatographic sample bottles. A total of 20 µL from each sample to be tested was mixed into a quality control (QC) sample. The remainder of each sample was used for LC–MS analysis. The sample extracts were analyzed in accordance with Table 1.

ESI–MS experiments were carried out on a Thermo Q-Exactive mass spectrometer, with spray voltages of 3.5 and −2.5 kV in positive and negative modes, respectively. The sheath and auxiliary gases were set to 30 and 10 arbitrary units, respectively. The capillary temperature was 325 °C. The analyzer scanned over a mass range of m/z 81-1000, for a full scan mode at a mass resolution of 70,000. Data-dependent acquisition (DDA) MS/MS experiments were performed with an HCD scan. The normalized collision energy was 30 eV. Dynamic exclusion was implemented to remove unnecessary information from the MS/MS spectra.

### 4.3. Metabolome Data Analysis

The raw LC–MS data were converted into mzXML format files by Proteowizard Data Analysis software (v3.0.8789) and subsequently processed peaks identification, filtration, and alignment via XCMS (www.bioconductor.org, accessed on 20 October 2020). XCMS’s default set, with the following changes: bw = 2, ppm = 15, peakwidth = c (5, 30), mzwid = 0.015, mzdiff = 0.01, and method = centWave. Each metabolite was confirmed based on their exact molecular weights, and the possible empirical formulae of the metabolites were speculated (molecular weight error < 20 ppm). The exact molecular weights were then used to identify potential biomarkers by confirmation in the Metlin (http://metlin.scripps.edu, accessed on 20 October 2020), massbank (http://www.massbank.jp/, accessed on 20 October 2020), Lipid Maps (http://www.lipidmaps.org, accessed on 20 October 2020), mzclound (https://www.mzcloud.org, accessed on 20 October 2020), and database built by Bionovogene Co., Ltd. (BioNovoGene, Suzhou, Jiangsu, China).

Qualitative and quantitative analyses of the metabolite data were log2-transformed for statistical analysis to improve normality. Metabolites from 30 samples were used for principal component analysis (PCA) and orthogonal partial least squares discriminant analysis (OPLS-DA), using R software to study metabolite accession-specific accumulation. The *p* and fold change values were set to 0.05 and 2.0, respectively. The Kyoto Encyclopedia of Genes and Genomes (KEGG) [19] database was used to study differential metabolites (*p* value < 0.01) in the 0 to 24 h samples.

### 4.4. RNA Extraction and RNA-Seq

Total RNA of the samples was extracted with an RNAprep Pure Plant kit (DP441, Tiangen, China). Illumina RNA-seq was performed by Metware Biotechnology Co., Ltd. (Wuhan, China). The RNA quality was determined with a NanoPhotometer spectrophotometer (IMPLEN, Munich, Free State of Bavaria, Germany), Qubit 2.0 fluorometer (Life Technologies, Carlsbad, CA, USA), and Agilent Bioanalyzer 2100 system (Agilent Technologies, Santa Clara, CA, USA). The poly(A) mRNA was enriched by magnetic beads with oligo(dT). The mRNA was randomly fragmented. First-strand cDNA was synthesized using the M-MuLV reverse transcriptase system. The RNA strand was then degraded by RNase H, and second-strand cDNA was synthesized using DNA polymerase. The double-stranded cDNAs were ligated to sequencing adapters. The cDNAs (~200 bp) were screened using AMPure XP beads. After amplification and purification, the cDNA libraries were obtained and sequenced using the Illumina Novaseq6000 system.

### 4.5. Sequence Data Processing

The raw reads were transformed by sequencing raw image data via CASAVA base recognition. To obtain high-quality data, adapters of the sequences were cut, and low-quality reads with ≥5 uncertain bases or over 50% Qphred ≤ 20 bases were removed using fastp. The GC content of the clean reads was calculated. The Q20 and Q30 values were also produced by Fast QC, in order to evaluate base quality. Then, the clean reads were mapped to the hickory reference genome using HISAT2 with the default parameters. Gene expression levels were determined using the RPKM (reads per kb per million reads) method.

### 4.6. Data Analysis

One-way analysis of variance was performed to highlight differences in relevant variables between treatments, followed by Tukey’s HSD post hoc multiple comparison tests. The normality and homogeneity of the variance of the data were assessed prior to analysis. All data analyses were carried out using the Statistical Package for Social Sciences program (SPSS 16.0, SPSS Inc., Chicago, Illinois, DE, USA). OriginPro 9.0 (Origin Lab Corporation, Northampton, MA, USA) was used for graphical presentation. All data are expressed as the means ± SD of three replicates.

## Figures and Tables

**Figure 1 molecules-27-03548-f001:**
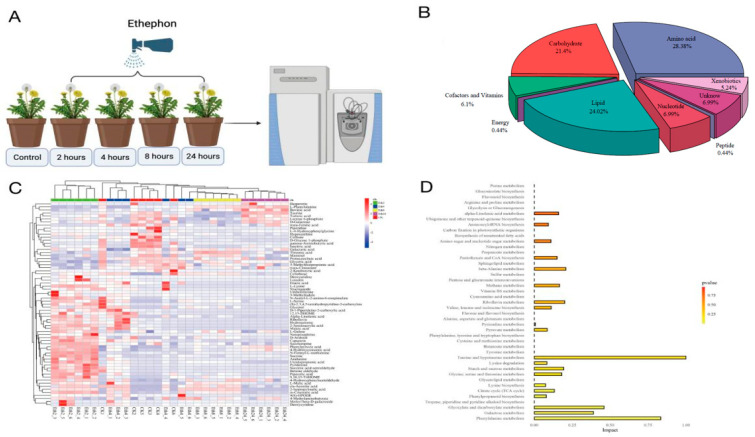
(**A**) Schematic diagram of the experimental design. (**B**) The identification and classification of TKS metabolites. (**C**) Clustering heatmap of the 5 different stages of TKS after exogenous ethephon spraying. (**D**) Overview of TKS pathway analysis.

**Figure 2 molecules-27-03548-f002:**
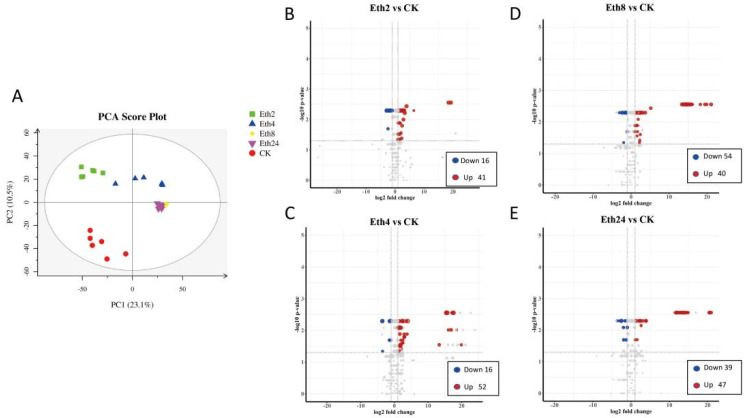
(**A**) PCA scores for the five TKS groups. (**B**–**E**) Volcano plots of the differentially accumulated metabolites in the Eth2 vs. CK, Eth4 vs. CK, Eth8 vs. CK, Eth24 vs. CK groups, respectively.

**Figure 3 molecules-27-03548-f003:**
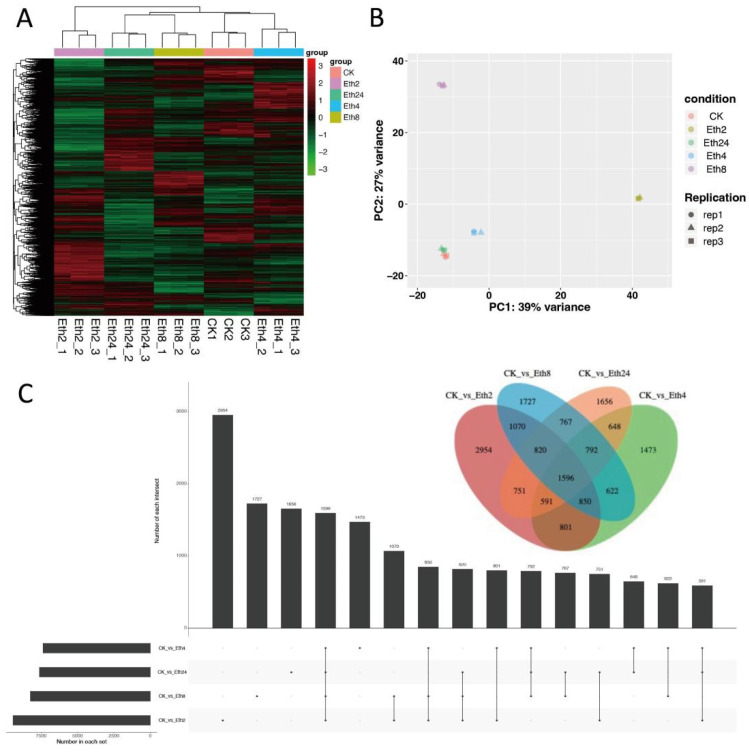
(**A**) Cluster analysis of the identified metabolites from the five stages. (**B**) Principal component analysis (PCA) of the five stages. (**C**) Advanced Venn diagram (UpSet) results for the transcriptome data from the four comparisons (CK vs. Eth2, CK vs. Eth4, CK vs. Eth8, and CK vs. Eth24).

**Figure 4 molecules-27-03548-f004:**
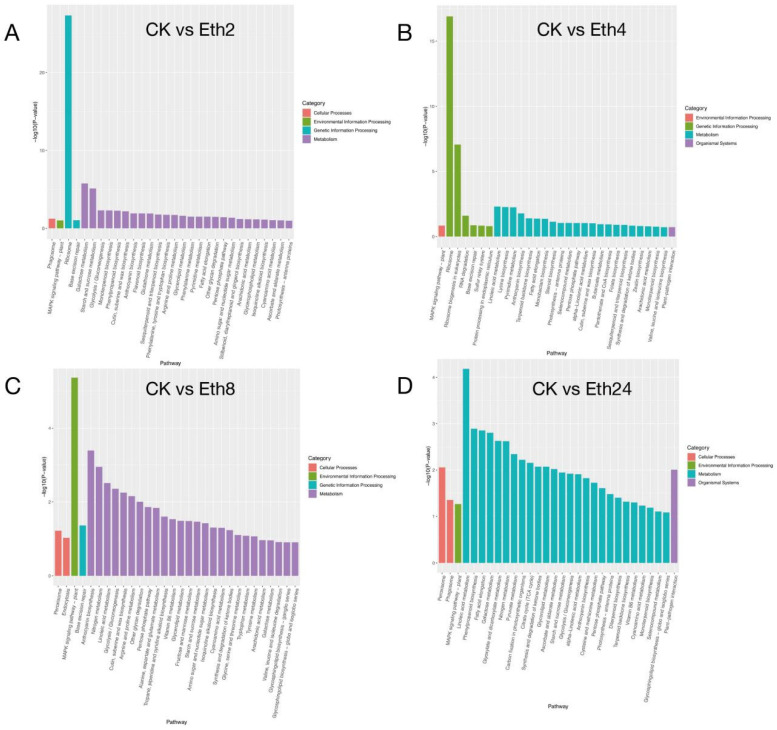
KEGG pathway classification map of the DEGs. (**A**) CK vs. Eth2; (**B**) CK vs. Eth4; (**C**) CK vs. Eth8; (**D**) CK vs. Eth24.

**Figure 5 molecules-27-03548-f005:**
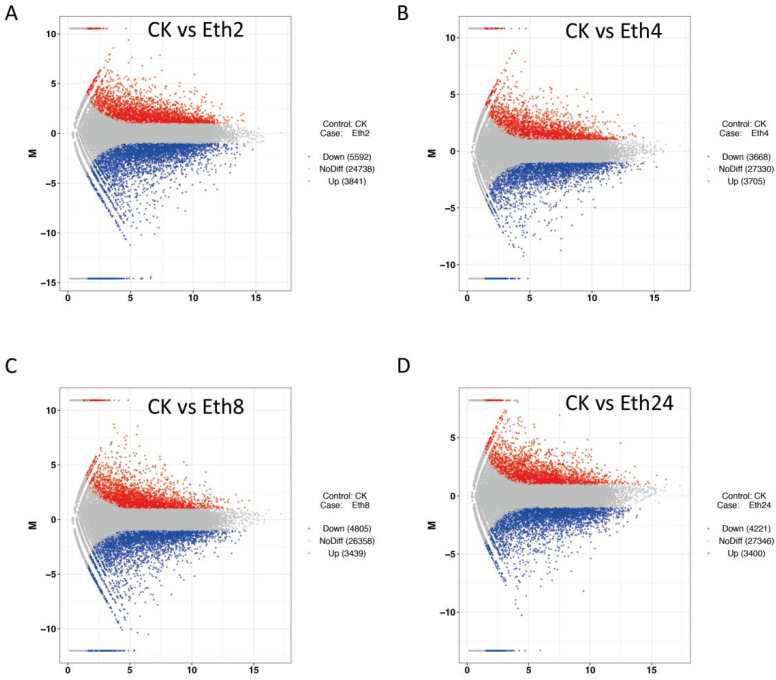
Volcano plots of differentially expressed genes in TKS. (**A**) CK vs. Eth2; (**B**) CK vs. Eth4; (**C**) CK vs. Eth8; (**D**) CK vs. Eth24.

**Figure 6 molecules-27-03548-f006:**
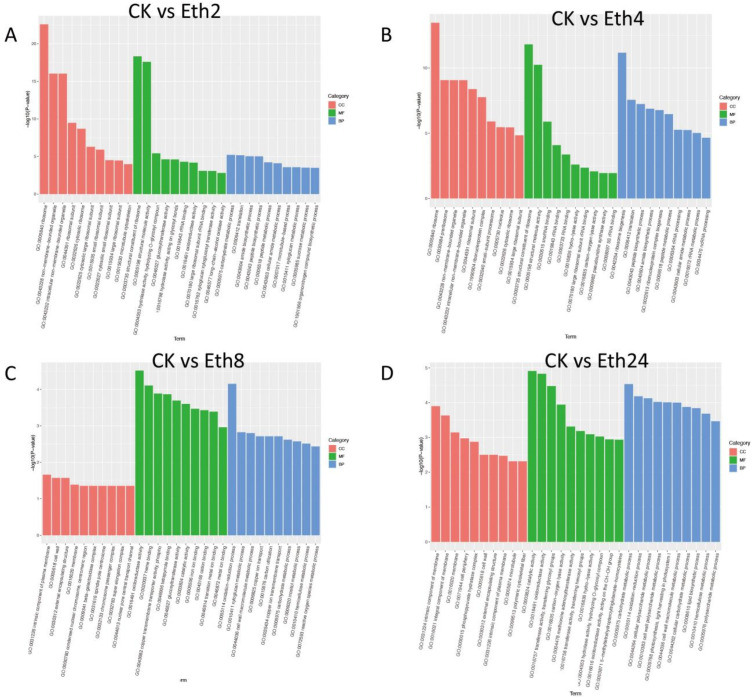
Gene ontology (GO) functional classifications of DEGs. (**A**) CK vs. Eth2; (**B**) CK vs. Eth4; (**C**) CK vs. Eth8; (**D**). CK vs. Eth24.

**Table 1 molecules-27-03548-t001:** Ultra-performance liquid chromatography analytical conditions.

Variables	Parameters
Column	ACQUITY UPLC^®^ HSS T3 (150 mm × 2.1 mm, 1.8 μm, Waters)
Solvent system	Mobile phase A (5 mM ammonium formate in water)
	Mobile phase B (5 mM ammonium formate in acetonitrile)
Gradient program	0~1 min, 2% B 1~9 min, 2~50% B 9~12 min, 50~98% B 12~13.5 min, 98% B 13.5~14 min, 98~2% B 14~17 min, 2% (B negative model)
Flow rate	0.25 mL/min
Column temperature	40 °C
Injection volume	2 μL

## Data Availability

The datasets presented in this study can be found in online repositories: https://www.ncbi.nlm.nih.gov/bioproject/PRJNA827459 (accessed on 16 April 2022).

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
