# Peer review of "Metabolic and Transcriptomic Analyses Reveal the Effects of Ethephon on Taraxacum kok-saghyz Rodin"

_molecules, 2022, doi:10.3390/molecules27113548_

Round 1
Reviewer 1 Report
The authors examined the primary and secondary metabolites of Taraxacum kok-saghyz (TKS) leaves using UPLC-ESI-MS / MS. In these leaves, they identified and characterized a total of 229 metabolites. Research results provide a better understanding of the main metabolites and their molecular mechanisms in TKS leaves. The article is very interesting and contains the results of new original research on Taraxacum kok-saghyz (TKS). The obtained results are of great scientific and practical importance and I believe that in the future TKS will be the subject of further research aimed at better use of the possibilities TKS.
The authors applied the correct research methodology and conducted appropriate UPLC-ESI/MS-MS analyzes. They performed the statistical analysis of the obtained results very well (PCA, PLS-DA). The inference is correct and the formulated conclusions are justified by the obtained results and the performed statistical analyzes.
My criticisms of the manuscript are as follows:
1. The authors use numerous acronyms and abbreviations for the names of various procedures and substances. Some of them are explained in the text, but many are not; e.g. NR, CK, KO, KOG, NT, PFAM. Despite the fact that some of these acronyms are known in the literature, I believe that the article should contain a list of acronyms with their explanations (in basic text or in supplementary materials), which will facilitate reading the text.
2. The authors included the Materials and Methods section in item 4, which is the last section in the substantive part of the manuscript. In order to improve the structure of the article, it would be better if this section was placed before the Results section and after the Introduction section. Alternatively, the Materials and Methods section could be included in the supplementary materials.
3. Figure 1 is composed of four figures A, B, C and D. Figure 1A is in fact a graphic abstract of the research methodology. Figures 1 C and 1 D are very illegible although they are very important throughout the publication. I propose that Figures 1 C and 1 D should be separate drawings. This could be dispensed with if the readers of the article had the opportunity to download and analyze these drawings as full-size drawings.
4. There is no graphic data in Figure 3B. The drawing is just blank. In my (PDF) version of the manuscript, I only see axes. This, of course, needs to be completed. Unless the authors have any data, in this situation they should remove Figure 3B.
5. Figure 5 A, B, C and D. Volcano plots of differential gene expression in TKS. A. CK vs. Eth2. B. CK vs. Eth4. C. CK vs. Eth8. D. CK vs. Eth24 has no graphical data. In the manuscript (PDF), I only see the coordinate axes. This drawing is very important to the value of work and should be completed.
6. In the title of the article, the authors used the term "ethylene spraying", while in the abstract and in the following text, the term "ethephon" was used. In my opinion, these are synonyms, however, the authors do not explain it in a short introduction. This can be confusing for an inexperienced reader. I propose that these terms should be standardized or explained.
Reviewer 2 Report
The authors of the manuscript by Zhang et al. describe the metabolic and transcriptomic changes upon ethylene spraying on Taraxacum plant leaves and its potential of increased rubber production in the roots or other unknown consequences. The data were presented in a descriptive manner and also the discussion lacks conclusions or sufficient clarification of agreement or disagreement of transcriptomics and metabolomics data. Are the transcriptomics data in line with the metabolomics data? Which metabolites might be suitable for dietary supplementation? Were these metabolites found in the roots in previous studies and which ones were most abundant in the roots compared to the leaves? Many figure panels were not well presented and should be improved.
Addressing the following main specific comments would highly improve the manuscript to allow a key message to the reader:
The extraction with methanol did not include the use of any internal standards (IS) to correct for extraction-based variations. It is described that a whole target IS mixture was added afterwards but not explained which one. Was the metabolomics analysis fully untargeted and the observed metabolites “only” features or putative metabolites that should be validated as the real metabolites? This should be clarified and described in more detail. The data analysis and processing to get to the putative metabolites based on full scan mass spectra should be described. The author only describe that dynamic exclusion was applied to the MS/MS spectra but there is no information what processing tools were used like AquireX or Compound Discoverer 3.2 or 3.3 from Thermo Fisher or any other?
Figure 1 is very blurry and the panel 1C, 1D cannot be read entirely. Please provide better quality figures. The schematic mass spectrometer shown in Figure 1 is not a Q Exactive (that was stated as used in the mansucript) and should be updated to avoid confusion.
The use of pooled QC in reference 14 (Sangster et al.2006) should not be cited for the extraction method as it describes just the advantage of the use of a pooled sample quality control (QC).
The reference of Benninghaus at al 2020, J Exp Bot. 2020 Feb 7; 71(4): 1278–1293. doi: 10.1093/jxb/erz512 should be included and discussed as this is a proteomics proteomics analysis of Taraxacum roots and might be valuable for correlations in the discussion
In Figure 2A, B: Why was a PLS-DA shown additionally to the PCA in panel A? There is no additional information here and PLS-DA are usually more robust and reliable when only two groups are compared and higher numbers of replicates were used.
In the PCA in Figure 3 B no data is shown in the graphics. Same is true for Figure 5 in the volcano plots.
Figure 4 and 6 are also very blurry in the labeling and not possible to be read properly.
Reviewer 3 Report
The manuscript report an interesting experimental approach on cultivated plant that can deserve promising applicative perspective.
In the present form tha manuscript highlight limitations and must be deeply revised. The plant material, treatments and all experimental procedures must be clearly defined because they should guarantee the experimental reproducibility.
The rationale base for the study should be deeper discussed. Presenting the data novelty just as "because was not yet tested" highlight the study as a instrumental virtuosism rather than as of scientific relevance.
Please accurately check the full plant name and authors.
L40 NR is for?
L52 "after 0 hours" is for time 0?
Par 4.1 should give all information to reproduce the experimental procedure. Please describe "samples", plant identification, type of cultivation, origin and preparation of the treatment, treatment procedures, leaves stages, plant identification, and all other relevant information.
The experimental groups, its numerosity and relative acronyms should be clearly defined
Can be usefull also to introduce a "control group"
Do authors also detect morphological or phenological variations?
The timing of sampling is consistent with mechanism of action of treatment ? please add comments
L319 50 Hz is for the homogenizer or for sonicator ? please clearly describes instrument and methodologies applied. Do the solvent was added after the sonication ?
It is not clear what are the SAMPLES and how many they are
Define what is a "detection bottle"
Define QC
L323 12000 rpm should be converted in g
The logical connection between ethephon (note that is a commercial product!) cited in material and method section and athylene cited in conclusion should be discussed
In the present version, all graphic and text of figures are illegible
Round 2
Reviewer 1 Report
The authors of the article significantly improved the manuscript and completed the missing elements. The content of the article is multidisciplinary, but I must emphasize that currently many scientific works are conducted on the border of various scientific disciplines. This enables the rapid development of science and allows for the effect of "added value", which I assess positively.
Reviewer 3 Report
Infinite variables can be applied to each experimental paradigm but usually the experimental design is based on rationale hypothesis. Probably nobody has investigate the effect of sprying degassed Coca Cola or aged white wine on plants, but I think that such experiment do not provides relevant innovativity and scientific soundnes only because it was never investigated. Explain why the described treatment is more pertinent could be a starting point to highlight the relevance of the study.
The researcher at the Institute of Plant Genetic Resources probably determined the plant where he collect the seeds and not the seeds itself.
L64: what is (K445) for?
Why authors used as control the "after 0 h" sample instead of time 0 (or untreated control group)? Do they exclude any "immedite" effects?
Please explain "grinder at 50 Hz". Is the frequency relevant for the grinding process? usually a different unit is used such as rpm.
L70: "The exogenous ethephon (100 μmol/L)" do authors used a chemical compound or commercial product? details such as chemical name, forms, origin, solvent for dilution and mode of preparation are missing.
Do authors quantified the amount of chemical treatment applied ?
L 88: Conditions or variables?
At least in the Discussion authors should comment the time-dependent conversion of etephon. Do authors consider that the effects should be attributed to the ethylene ?
